# Characteristics of antimicrobial resistance in *Escherichia coli* isolated from retail meat products in North Carolina

Mabel Kamweli Aworh[1], Siddhartha Thakur[1], Catherine Gensler[2], Erin Harrell[1], Lyndy Harden[1], Paula J. Fedorka-Cray[1], Megan Jacob[1]*

**1** Department of Population Health and Pathobiology, College of Veterinary Medicine, North Carolina State University, Raleigh, North Carolina, United States of America, **2** Department of Agricultural and Human Sciences, College of Agriculture and Life Sciences, North Carolina State University, Raleigh, North Carolina, United States of America

* mejacob@ncsu.edu

**Data Availability Statement:** All relevant data are within the manuscript and its Supporting Information files.

## Abstract

### Background

*Escherichia coli* is commonly used as an indicator for antimicrobial resistance (AMR) in food, animal, environment, and human surveillance systems. Our study aimed to characterize AMR in *E. coli* isolated from retail meat purchased from grocery stores in North Carolina, USA as part of the National Antimicrobial Resistance Monitoring System (NARMS).

### Materials and methods

Retail chicken (breast, n = 96; giblets, n = 24), turkey (n = 96), and pork (n = 96) products were purchased monthly from different counties in North Carolina during 2022. Label claims on packages regarding antibiotic use were recorded at collection. *E. coli* was isolated from meat samples using culture-based methods and isolates were characterized for antimicrobial resistance using whole genome sequencing. Multi-locus sequence typing, phylogroups, and a single nucleotide polymorphism (SNP)-based maximum-likelihood phylogenic tree was generated. Data were analyzed statistically to determine differences between antibiotic use claims and meat type.

### Results

Of 312 retail meat samples, 138 (44.2%) were positive for *E. coli*, with turkey (78/138; 56.5%) demonstrating the highest prevalence. Prevalence was lower in chicken (41/138; 29.7%) and pork (19/138;13.8%). Quality sequence data was available from 84.8% (117/138) of the *E. coli* isolates, which included 72 (61.5%) from turkey, 27 (23.1%) from chicken breast, and 18 (15.4%) from pork. Genes associated with AMR were detected in 77.8% (91/117) of the isolates and 35.9% (42/117) were defined as multidrug resistant (MDR: being resistant to ≥3 distinct classes of antimicrobials). Commonly observed AMR genes included *tetB* (35%), *tetA* (24.8%), *aph(3'')-lb* (24.8%), and *bla*TEM-1 (20.5%), the majority of which originated from turkey isolates. Antibiotics use claims had no statistical effect on MDR *E. coli* isolates from the different meat types ($X^2$ = 2.21, p = 0.33). MDR was observed in

**Funding:** FDA National Antimicrobial Resistance Monitoring System – ST Grant number 1U01FD007145-01 and GenomeTrakr program – Grant number 1U18FD00678801. The funders had no role in study design, data collection and analysis, decision to publish, or preparation of the manuscript.

**Competing interests:** The authors have declared that no competing interests exist.

isolates from meat products with labels indicating "no claims" (n = 29; 69%), "no antibiotics ever" (n = 9; 21.4%), and "organic" (n = 4; 9.5%). Thirty-four different replicon types were observed. AMR genes were carried on plasmids in 17 *E. coli* isolates, of which 15 (88.2%) were from turkey and two (11.8%) from chicken. Known sequence types (STs) were described for 81 *E. coli* isolates, with ST117 (8.5%), ST297 (5.1%), and ST58 (3.4%) being the most prevalent across retail meat types. The most prevalent phylogroups were B1 (29.1%) and A (28.2%). Five clonal patterns were detected among isolates.

## Conclusions

*E. coli* prevalence and the presence of AMR and MDR were highest in turkey retail meat. The lack of an association between MDR *E. coli* in retail meat and antibiotic use claim, including those with no indication of antimicrobial use, suggests that additional research is required to understand the origin of resistance. The presence of ST117, an emerging human pathogen, warrants further surveillance. The isolates were distinctly diverse suggesting an instability in population dynamics.

## Introduction

Surveillance of antimicrobial resistance (AMR) along the one-heath continuum, including in the food chain, is necessary to reduce the impact of resistant bacteria on health and prioritize policies and areas of intervention. The value and importance of one-health surveillance systems have been well described [1]. *Escherichia coli*, a gram-negative bacteria common to the human and animal gastrointestinal tract, is often used as an indicator organism in AMR surveillance systems [2]. Studies have shown that food products are potential reservoirs of pathogenic *E. coli* for humans [3, 4]. Most foodborne outbreaks in humans caused by *E. coli* have been associated with the consumption of contaminated food products of animal origin or contaminated with animal feces [5]; additionally, the role of these organisms in transmission of AMR to naturally-occurring human strains remains unknown and a concern. *E. coli* have flexible fitness mechanisms, making them easily adaptable to environmental conditions [6]. They can persist on surfaces for long periods of time and can be isolated from ready-to-eat food products [7].

To track AMR in food products of animal origin in the United States, the National Antimicrobial Resistance Monitoring System (NARMS) was established in 1996. The NARMS program which is operated by the Centers for Disease Control and Prevention (CDC), Food and Drug Administration (FDA) and U.S. Department of Agriculture (USDA) routinely conducts surveillance of foodborne pathogens in humans, retail meat products and food animals respectively. *E. coli* is one of the indicator bacteria identified for AMR surveillance in NARMS [8]. The NARMS program contributes to the promotion and protection of public health by disseminating knowledge regarding new bacterial resistance, the distinction between resistant and susceptible illnesses, and the effects of treatments meant to stop the spread of resistance [9]. Since its inception, the NARMS program has been solely responsible for continued tracking of antimicrobial resistance among human and animal related populations through the food supply.

The use of antimicrobials in any setting has potential implications on the development and maintenance of AMR. In the United States, retail food products including meat have been marketed with antibiotic use claims since the mid-2000s [10]. There are public concerns that

the use of antimicrobials in food producing animals may affect the efficacy of similar drugs in human medicine especially through selection pressure of resistant bacteria and their ability to be transferred to humans through the food chain [11, 12]. The scientific evidence to support the risk of food products in disseminating AMR bacteria, including *E. coli*, is less clear. Studies have shown that resistance genes in bacteria such as *E. coli* can be transmitted at the human-animal-environment interface [13, 14]. Increasingly, modern molecular methods such as whole-genome sequencing (WGS) have been used to detect and trace the presence of resistance genes, mobile genetic elements and plasmids in *E. coli*, including from retail meat [15]. The NARMS surveillance program uses a WGS-based method for evaluating AMR in *E. coli* isolates from retail meat products.

Historically, the most prevalent AMR genes detected in *E. coli* from retail meat from the NARMS surveillance program include genes associated with erythromycin (*mph*(A)), tetracycline (*tet*(A), *tet*(B) and *tet*(C)), sulfonamide (*sul1* and *sul2*) and plasmid mediated quinolone (*qnr; gyr* or *par* mutations) resistance [16–18]. Recent analysis of the NARMS Genome Trackr database showed that common phylogroups detected in *E. coli* recovered from retail meat were A, B1, B2, C, Clade I, D, E, F, and G [18]. Previous NARMS studies have reported *E. coli* prevalence of 47.5% in all retail meat products, with higher prevalence (90.7%) reported in turkey products [17, 19]. While one study reports more than 50% of the isolates demonstrated multidrug resistance (MDR), which showed turkey meat-derived *E. coli* isolates with the highest resistance other studies have reported a lower MDR prevalence of 14.3% [17, 19].

Here, we provide updated prevalence estimates and characterization of the common AMR mechanisms and phylogeny characteristics of *E. coli* isolated from retail meat with different antimicrobial use claims including chicken breasts, pork chops and ground turkey purchased from retail grocery stores in North Carolina, USA.

## Materials and methods

### Retail meat sampling

Between January and December 2022, 312 meat products comprised of retail chicken cuts (breast; bone-in/skin-on; n = 96), ground turkey (n = 96), pork chop (n = 96) and chicken giblets (liver, gizzard, or heart; n = 24) were purchased from grocery supermarkets across five municipalities and eight counties in North Carolina in accordance with NARMS project protocol [20]. Monthly, 26 samples were randomly selected and purchased from grocery stores using zip codes that geographically represented each location based on US FDA assignment. At each sampling, eight chicken breast, eight pork chop, eight ground turkey and one package each of chicken heart, liver or gizzards were purchased.

As per the NARMS protocol for 2022, the demographic information collected for each purchased meat sample included antibiotic use claims apparent on product label, store name, store location, brand name, sell-by-date, purchase date, lab processing date and season of retail meat purchase (winter, spring, summer, and fall). Winter is considered December, January, and February; spring is March through May; summer is June through August; and fall is September through November. The samples were kept on ice in an isothermal container, during transportation from the grocery stores to the laboratory. On arrival at the lab, samples were refrigerated at 4°C and processed within 96 hours for detection of multiple bacteria, including *E. coli*.

### Bacterial culture

We followed the provided NARMS 2022 retail meat surveillance laboratory protocol; briefly, this began with 50g of each sample aseptically cut into a stomacher bag where 250 ml of Buffer

Peptone Water (Thermo Fisher Scientific, Waltham, MA) was added, and the sample was homogenized on a shaker at 200rpm for 15 mins. After mixing, the sample was placed into a sterile, plastic container (Fisherbrand™) with 50 ml of double strength (2X) MacConkey broth (Thermo Fisher Scientific, Waltham, MA), mixed well and incubated at 35˚C for 24h. Following incubation, a 10ul loopful of enrichment broth was plated onto MacConkey agar (BD BBL™, Sparks, MD) and incubated at 35˚C for 24h. One colony demonstrating typical *E. coli* phenotypic morphology (pink, round) was picked from each plate and streaked for isolation onto blood agar plates (Thermo Fisher Scientific, Lenexa, KS) which were incubated at 35˚C for 18-24h. Indole (BD BBL™, Sparks, MD) and oxidase (BD BBL™, Sparks, MD) quick tests were performed on suspected *E. coli* colonies; indole-positive, oxidase-negative colonies were confirmed to be *E. coli* by matrix-assisted laser desorption/ionization-time of flight (MALDI TOF; BioMerieux). All confirmed *E. coli* isolates were saved in cryovials containing *Brucella* broth (Thermo Fisher Scientific, Lenexa, KS) with 15% glycerol and subsequently stored at -80˚C for sequencing analysis.

## DNA extraction and whole genome sequencing

DNA was extracted from each *E. coli* isolate using a modified version of the Qiagen DNeasy PowerLyzer microbial kit (Qiagen, Hilden, Germany). Using a 10μl loop, four passes of cells were added to 300μl of PowerBead solution in a 1.5 ml microcentrifuge tube. Next, 50μl of solution-SL was added to the PowerBead tubes which were immediately vortexed for 5 seconds to re-suspend. The bacterial suspension (300μl) was transferred to the PowerBead tubes which were placed in a bead mill (speed = 6; time = 1min) to homogenize. The PowerBead tubes were centrifuged (30secs at 17,000xg), and the supernatant was transferred to new 1.5 ml microcentrifuge tube (Qiagen, Hilden, Germany). Afterwards 100μl of solution-IRS was added to the supernatant and vortexed briefly, followed by a 5 min incubation at 4˚C. After incubation the tubes were centrifuged, and the supernatant again transferred to a new 1.5 ml microcentrifuge tube each containing 900μl of solution-SB. The tubes were vortexed and 650μl of supernatant transferred to the spin column for spin washing (3 washes) and final elution. The Nanodrop 2000 Spectrophotometer was used to conduct a quality check for all the DNA samples. The ratio of absorbance at 260 nm and 280 nm is used to assess the purity of DNA. A ratio of ~1.8 is generally accepted as "pure" for DNA. Subsequently the DNA concentration was quantified using Qubit 4.0 Fluorometer (ThermoFisher Scientific, Waltham, MA). DNA libraries of each sample were prepared for whole genome sequencing (WGS) using a Nextera XT kit (Illumina, San Diego, CA). Briefly, 0.3 ng/μl of DNA from each *E. coli* isolate was pooled together and sequenced on an Illumina MiSeq platform (Illumina, San Diego, CA) using 2 x 250 or 2 x 300 paired-end approach. The raw paired end reads from the sequencer were demultiplexed and submitted to the NCBI database where they were assigned an accession number [21].

## Assembly and assessment of genes

*E. coli* genome sequences were assembled *de novo* using SPAdes version 3.15.4 via a web-based genome assembly service provided by the Bacterial and Viral Bioinformatics Resource Center accessed online at https://www.bv-brc.org/app/Assembly2. The *in silico* analysis of acquired resistance genes and replicon typing for each *E. coli* isolate was used to identify AMR genes harbored on plasmid replicons and was conducted using the Mobile Element Finder tool (database version 1.0.2, 2020-06-09) accessed online via the Center for Genomic Epidemiology (CGE) website (https://cge.food.dtu.dk/services/MobileElementFinder/) [22]. From the output generated, if an isolate had at least one gene from either of the following sets: (i) *ompT* and

*hlyF*, and (ii) sitABCD, it was deemed ColV plasmid positive [23]. The Mobile Element Finder tool interfaced with the ResFinder 4.1 tool to match individual genes for each *E. coli* isolate to an annotated resistance gene using a 90–100% identity, 60% minimum length, and 90% threshold [22]. PlasmidFinder 2.1 tool (database version 2023-01-18) set at a 95% minimum identity and 60% coverage was used for replicon typing. The CGE pMLST 2.0 tool (database version 2023-04-24) set at a 95% minimum identity and based on IncF-RST configuration was used for *in silico* plasmid MLST typing to determine the plasmid replicon sequence types [24]. We defined molecular MDR as the presence of resistance genes conferring AMR to three or more distinct classes of antimicrobials in the database.

### Multi-locus sequence typing (MLST) of *E. coli* isolates

*In silico* prediction of MLST was performed by submitting the assembled genome of an isolate to the *E. coli* PubMLST database (https://pubmed.ncbi.nlm.nih.gov/30345391/). The MLST 2.0 (2022-11-14) tool on CGE website analyzed the contigs using previously described schemes by Achtman to assign sequence types (STs) based on allelic variations amongst seven housekeeping genes (*adk*, *fumC*, *gyrB*, *icd*, *mdh*, *purA*, and *recA*) [25]. *E. coli* isolates with identical sequences at all seven loci were assigned STs however those without perfect matches were usually identified as novel or unknown.

### Determination of *E. coli* phylogroups and phylogeny

The phylogenetic classification of the *E. coli* genomes was conducted using *in silico* Clermon-Typing 1.4.1 tool as previously described [26]. The ClermonTyper web interface is freely accessible at http://clermontyping.iame-research. center/. The Nextflow workflow was used to map the *E. coli* whole genome fastq pair end reads to a reference genome as well as call variants which were used to generate a maximum likelihood phylogenetic tree (https://gitlab.com/cgps/ghru/pipelines/snp_phylogeny). The clonal relationship between isolates was estimated using a pairwise single nucleotide polymorphism (SNP) analysis. The SNP analysis was used to determine how closely related the isolates were using 'snp-dists', a command line bioinformatics tool for transforming multiple DNA sequence alignment into a distance matrix (https://github.com/tseemann/snp-dists). Isolates that were less than 30 SNPs apart were related. The maximum likelihood phylogenetic tree was visualized using the interactive Tree of Life tool–iTOL version 6 (http://itol.embl.de/itol.cgi).

### Data analyses

Data were analyzed using R version 4.3 statistical software. The data were summarized by calculating frequencies and proportions. Pearson's chi-squared test and Fisher's exact test were used to compare MDR prevalence across retail meat types and significance was determined at a p-value $\leq$ 0.05. The additional data file for this study contains a list of accession numbers for individual Sequence Read Archive (SRA) for the *E. coli* isolates (S1 File).

## Results

From January through December 2022, a total of 312 meat samples were purchased from 48 grocery stores in North Carolina, representing 8 counties. Of these samples, 69 (22.1%) were labeled with "no-antibiotic-ever", 39 (12.5%) were labeled as organic, and 204 (65.3%) were identified as having no antibiotic use claim present. Of the 312 samples, 138 were positive for *E. coli* with an overall prevalence of 44.2%. Of these, 78 (56.5%) *E. coli* isolates were recovered from ground turkey, followed by chicken breast (n = 27, 19.6%), chicken giblets (n = 14,

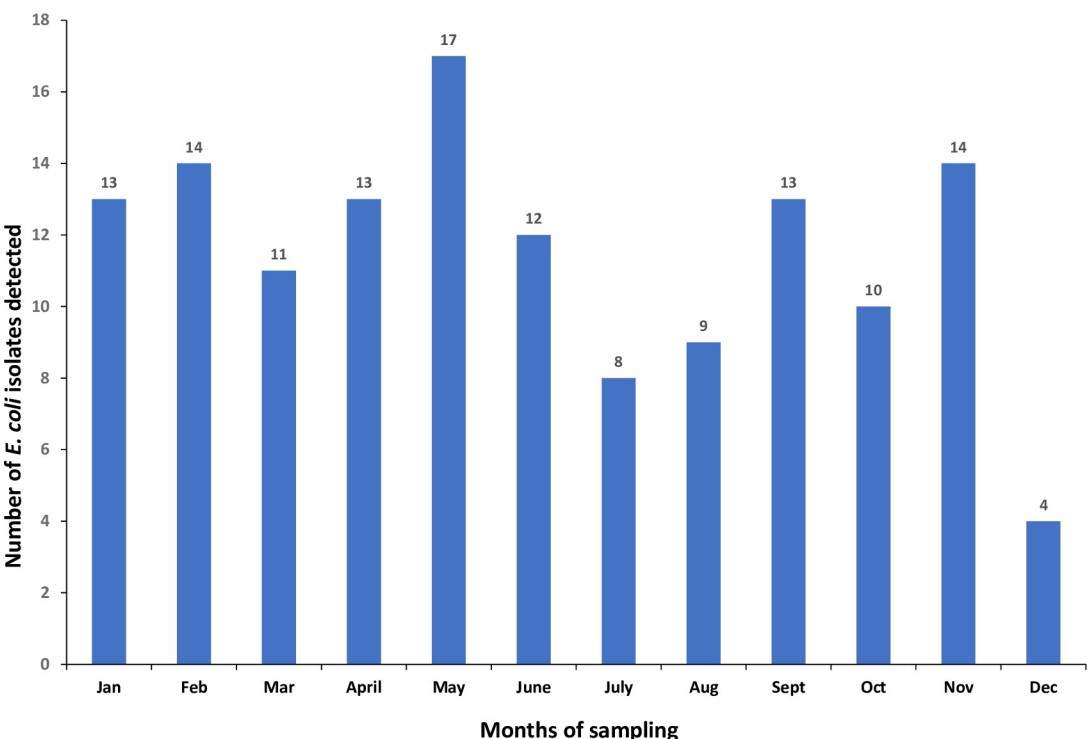

**Fig 1. *Escherichia coli* occurrence in retail meat by month of sampling in North Carolina.** The highest *E. coli* occurrence was observed in May and the lowest in December 2022.

10.1%) and pork chops (n = 19, 13.8%). The differences observed in prevalence of *E. coli* among retail meat types were statistically significant ($X^2$ = 13.17; p<0.01). Regardless of meat type, the distribution of *E. coli* isolates across meat types was 15.2% (n = 21) in organic products, 18.8% (n = 26) in "no-antibiotic-ever" products and 65.9% (n = 91) in products with "no claims". The observed prevalence difference was not statistically significant across label claims ($X^2$ = 2.67; p = 0.26). The prevalence of *E. coli* isolates differed by month of sampling with the highest prevalence observed in May (17/26; 65.3%), followed by February and November (14/26; 53.8% each) while the least prevalence was observed in December (4/26; 15.3%) as shown in Fig 1. This observed difference between months was statistically significant ($X^2$ = 19.29; p = 0.05). However, there was no significant association between *E. coli* prevalence and season of retail meat purchase from 138 positive isolates ($X^2$ = 4.73; p = 0.19). The highest prevalence (41/78; 52.6%) was observed in Spring and the lowest prevalence (29/78; 37.2%) was observed in Summer.

Of all *E. coli* isolates, 84.8% (117/138) had good quality sequence data available for further characterization. The accession numbers for 117 paired end reads for the NARMS *E. coli* isolates collected in North Carolina by the North Carolina State University (NCSU) College of Veterinary Medicine (CVM) Thakur Molecular Epidemiology Laboratory for 2022 have been uploaded onto the National Center for Biotechnology Information (NCBI) database https://www.ncbi.nlm.nih.gov under bio projects accession number PRJNA292663. Of these, 72 (61.5%) were recovered from ground turkey, 27 (23.1%) from chicken breast and 18 (15.4%) from pork chops. AMR genes were detected in 77.8% (91/117) of isolates, and 35.9% (42/117) were defined as MDR. The distribution of MDR *E. coli* isolates across retail meat type was 83.3% (35/42) for ground turkey, 9.5% (4/42) from chicken breast and 7.1% (3/42) from pork

**Table 1. Antimicrobial resistance (AMR) genes detected in *Escherichia coli* isolated from retail meat in North Carolina, USA.**

| AMR genes | AMR class | Overall n = 117 (%) | Ground turkey n = 72 (%) | Chicken breast n = 27 (%) | Pork chop n = 18 (%) |
|---|---|---|---|---|---|
| *aac(3)-lld* | Aminoglycosides | 3 (2.6) | 3 (4.2) | 0 (0) | 0 (0) |
| *aac(3)-IV* | | 15 (12.8) | 14 (19.4) | 1 (3.7) | 0 (0) |
| *aac(3)-Vla* | | 5 (4.3) | 3 (4.2) | 2 (7.4) | 0 (0) |
| *aadA1* | | 15 (12.8) | 9 (12.5) | 3 (11.1) | 3 (16.7) |
| *aph(3")-lb* | | 29 (24.8) | 24 (33.3) | 2 (7.4) | 3 (16.7) |
| *aph(4)-la* | | 15 (12.8) | 15 (20.8) | 0 (0) | 0 (0) |
| *aph(6)-ld* | | 25 (21.4) | 20 (27.8) | 2 (7.4) | 3 (16.7) |
| *bla*CARB-2 | Beta-lactamases | 1 (0.9) | 1 (1.4) | 0 (0) | 0 (0) |
| *bla*HERA-3 | | 9 (7.7) | 9 (12.5) | 0 (0) | 0 (0) |
| *bla*OXA-1 | | 1 (0.9) | 1 (1.4) | 0 (0) | 0 (0) |
| *bla*TEM-1 | | 24 (20.5) | 20 (27.8) | 2 (7.4) | 2 (11.1) |
| *bla*TEM-141 | | 1 (0.9) | 1 (1.4) | 0 (0) | 0 (0) |
| *bla*TEM-206 | | 1 (0.9) | 1 (1.4) | 0 (0) | 0 (0) |
| *dfrA1* | Folate pathway antagonists | 2 (1.7) | 1 (1.4) | 1 (3.7) | 0 (0) |
| *dfrA14* | | 1 (0.9) | 1 (1.4) | 0 (0) | 0 (0) |
| *dfrA15* | | 1 (0.9) | 0 (0) | 0 (0) | 1 (5.6) |
| *sul1* | | 11 (9.4) | 8 (11.1) | 2 (7.4) | 1 (5.6) |
| *sul2* | | 14 (12.0) | 11 (15.3) | 3 (11.1) | 0 (0) |
| *floR* | Phenicols | 1 (0.9) | 1 (1.4) | 0 (0) | 0 (0) |
| *fosA7* | Phosphonic antibiotics | 1 (0.9) | 1 (1.4) | 0 (0) | 0 (0) |
| *qacE* | Quinolones | 8 (6.8) | 5 (6.9) | 2 (7.4) | 1 (5.6) |
| *qacL* | | 2 (1.7) | 2 (2.8) | 0 (0) | 0 (0) |
| *qnrB19* | | 2 (1.7) | 2 (2.8) | 0 (0) | 0 (0) |
| *qnrS1* | | 1 (0.9) | 1 (1.4) | 0 (0) | 0 (0) |
| *Inu(F)* | Macrolides | 2 (1.7) | 0 (0) | 0 (0) | 2 (11.1) |
| *tetA* | Tetracyclines | 29 (24.8) | 23 (31.9) | 5 (18.5) | 1 (5.6) |
| *tetB* | | 41 (35.0) | 36 (50.0) | 2 (7.4) | 3 (16.7) |
| *tet* gene (A&B) | | 60 (51.3) | 49 (68.1) | 7 (25.9) | 4 (22.2) |

chop, respectively. Sequencing analysis of AMR genes showed that the most prevalent resistance genes detected in the *E. coli* isolates from retail meats belonged to aminoglycosides (91.5%; 107/117), tetracyclines (59.8%; 70/117), beta-lactamases (31.6%; 37/117), folate pathway antagonists (25.2%; 29/117) and quinolones (11.1%; 13/117) as shown in Table 1.

Within each retail meat type, we assessed the impact of label antibiotic use disclosure as displayed on the packaging on MDR. The different label types did not statistically impact the resistance of MDR *E. coli* isolates (p = 0.33). MDR in turkey *E. coli* isolates was significantly different when compared to all other retail meat types regardless of the label claim (p < 0.01) (Table 2).

The MLST analysis showed that the 117 *E. coli* isolates belonged to 81 known sequence types (STs) with the most prevalent being ST117 (8.5%;10/117), ST297 (5.1%; 6/117), ST58 (n = 4; 3.4%), and three isolates each (2.6%) for ST10, ST126, ST602, and ST1079. Numerous other STs were identified among the *E. coli* isolates with one or two isolates represented; ST131 and ST371 were not detected. The most prevalent MLST types detected in the isolates are shown in Fig 2.

The 117 isolates in this study belonged to nine different phylogroups with the most prevalent phylogroup representing B1 (n = 34; 29.1%) followed by phylogroup A (n = 33; 28.2%), phylogroup B2 (n = 16; 13.7%) and phylogroup G (n = 11; 9.4%). Isolates with phylogroup B1

**Table 2. Association of multidrug resistance (MDR) *Escherichia coli* in retail meat with antibiotics use label claims.**

| Variables | | MDR n = 42 (%) | Not MDR n = 75 (%) | Pearson's chi-squared | p-value |
|---|---|---|---|---|---|
| **Antibiotics use claims on package (all meat types)** | No claim on package | 29 (69.0) | 50 (66.7) | 2.21 | 0.33 0.34* |
| | Organic | 4 (9.5) | 14 (18.7) | | |
| | No antibiotics use ever | 9 (21.4) | 11 (14.7) | | |
| **Retail Meat types** | Ground Turkey | 35 (83.3) | 37 (49.3) | 13.17 | ≤ **0.01** < **0.01*** |
| | Chicken breast | 4 (9.5) | 23 (30.7) | | |
| | Pork chop | 3 (7.1) | 15 (20.0) | | |
| **#Ground Turkey (antibiotics use claims)** | No claim on package | 26 (61.9) | 25 (33.3) | 4.39 | 0.11 0.12* |
| | Organic | 2 (4.8) | 8 (10.7) | | |
| | No antibiotics use ever | 7 (16.7) | 4 (5.3) | | |
| **##Ground Turkey (antibiotics use claims)** | No claim on package | 26 (61.9) | 25 (33.3) | 0.14 | 0.71 0.61* |
| | Claim (Organic/ "no antibiotics ever") | 9 (21.4) | 12 (16) | | |

*Fisher's Exact Test; #Analysis of different antibiotics use claims among ground turkey meat products; ##The second analysis compares the difference between "no claim" versus "claim" among ground turkey meat products.

were recovered from ground turkey (19/34; 55.9%), chicken breast (8/34; 23.5%) and pork chop (7/34; 20.6%) (Fig 3). Most isolates assigned ST297 (n = 4) and ST58 (n = 2) belonged to phylogroup A, while half of the isolates assigned ST117 (n = 5) belonged to phylogroup B1.

Thirty-four different replicon types were observed among *E. coli* isolates from retail meat. The most prevalent replicon types were IncFIB(AP001918) (n = 73, 62.4%), Col(MG828) (n = 50, 42.4%) and IncFIC(FII) (n = 43, 36.4%) as shown in Table 3. Thirteen replicon types

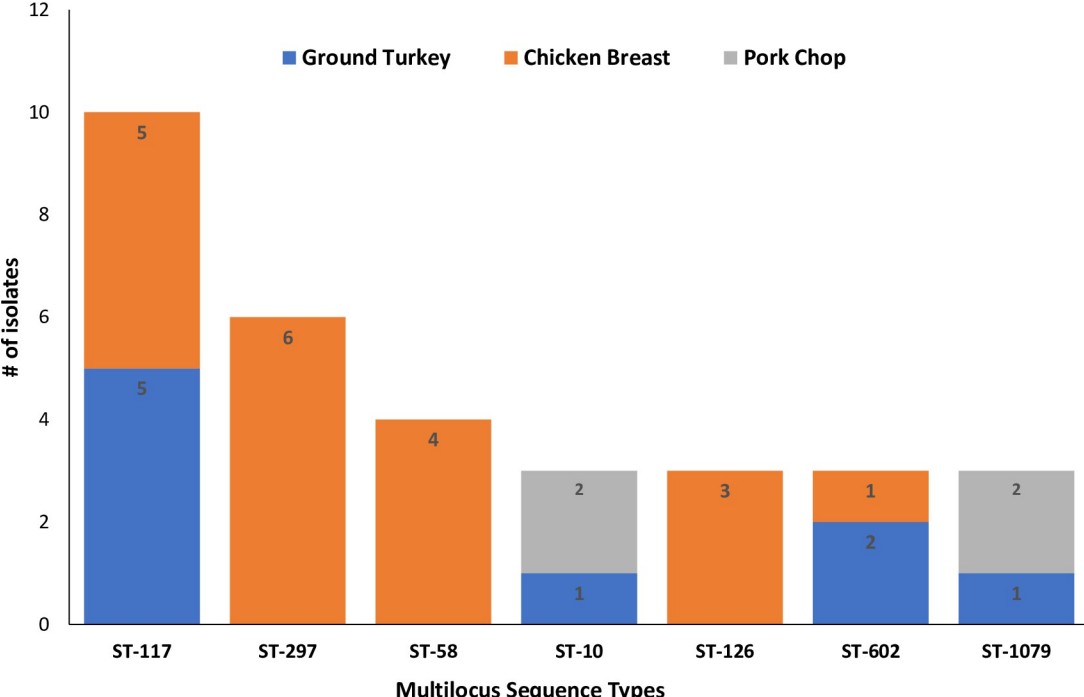

**Fig 2. Multi-locus sequence types of *Escherichia coli* isolated from ground turkey, chicken breast, and pork chops in North Carolina, USA.** Each bar represents the various *E. coli* sequence types for isolates obtained from the different retail meat sources.

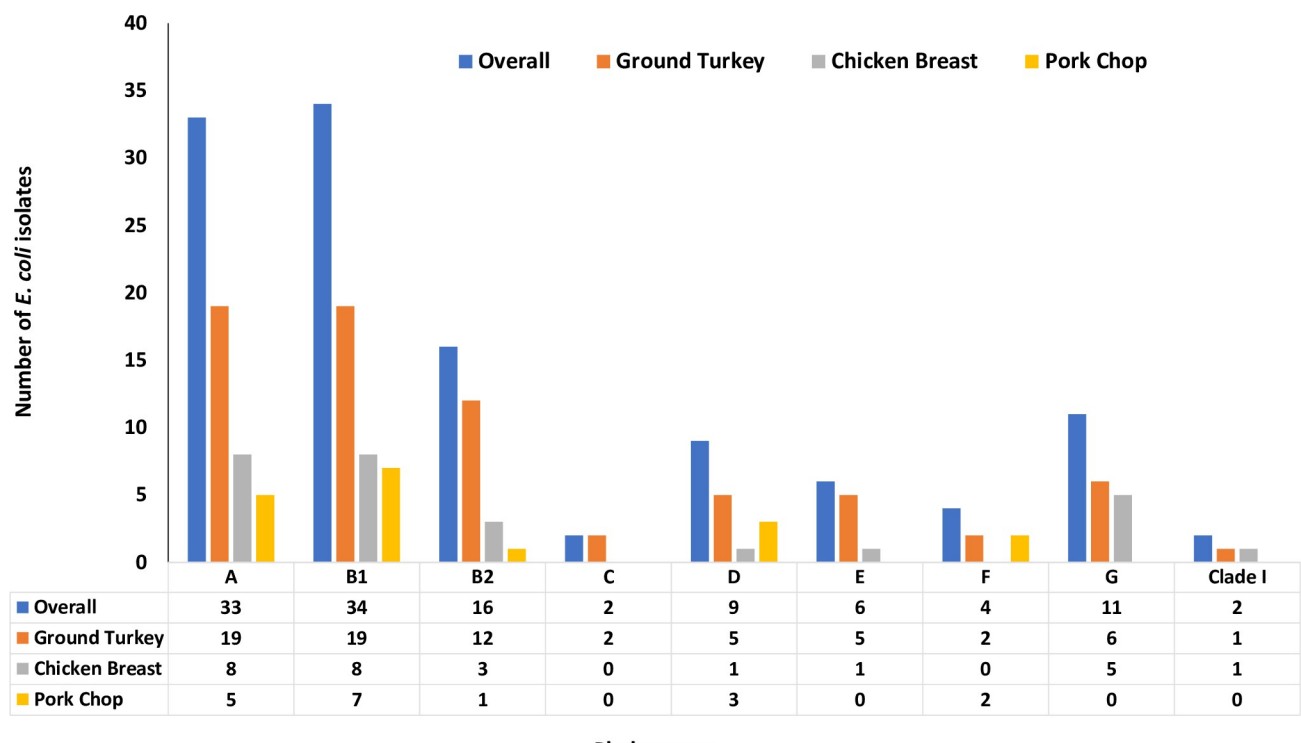

**Fig 3. Phylogenetic classification of *Escherichia coli* isolates from retail meat types in North Carolina, 2022.**

were common to isolates from all retail meat sources: IncFIB(AP001918), IncFII, IncFIC(FII), IncFII(29), Col(MG828), Col156, ColpVC, Col440I, IncX1, IncFII(pCoo), IncFIA, IncI1, and p0111. IncFIB(AP001918) was the most prevalent in all meat types: ground turkey (n = 51), chicken breast (n = 18) and pork chop (n = 3) followed by Col(MG828) detected in ground turkey (n = 35), chicken breast (n = 12) and pork chop (n = 3).

Of 117 *E. coli* isolates, only 92 (78.6%) had 50 known F plasmid replicon sequence types. The most common were F18:A-:B1 (12/92, 13%); F18:A-:B- (6/92, 6.5%); F4:A-:B20 (6/92, 6.5%); F18:A27:B1 (5/92, 5.4%); F24:A-:B1 (4/92, 4.3%); F18:A-:B20 (3/92, 3.3%); F2:A-:B- (3/92, 3.3%) and F4:A-:B1 (3/92, 3.3%) as shown in S1 Table. The F18:A-:B1 sequence type originated mostly from turkey (8/12) and chicken (4/12) meat products. Eighteen of the isolates did not have any F plasmid replicons while the remaining seven isolates were not typable due to the presence of novel alleles. The prevalent F plasmid replicon sequence types observed in *E. coli* ST117 (n = 10) were F18:A-:B1 (4/10) of which most were from chicken (3/4) and F24:A-:B1 (3/10) from ground turkey (2/3). Plasmid profiling identified presumptive ColV positive plasmid replicons in 72.6% (85/117) of *E. coli* isolates recovered from different retail meat products (S2 Table). Of these, 67.1% (57/85) were recovered from ground turkey, chicken 25.9% (22/85) and pork 7.1% (6/85).

Overall, 117 isolates were used to construct a SNP-based maximum likelihood phylogenetic tree (Fig 4). The *E. coli* isolates from retail meat were very diverse and only isolates from the same meat type were clonally related. More than half (57.3%) of the isolates were clustered into two main phylogroups and seven different STs. The SNPs matrix based on the core genome of 117 *E. coli* strains showed five clonal relationships with a pairwise SNP difference of below 30 (Table 4). *E. coli* isolates with a clonal relationship were clustered together based on phylogroups and STs as shown on the phylogenetic tree in Fig 4.

**Table 3. Plasmid replicon types detected in *Escherichia coli* isolates from retail meat types in North Carolina, 2022.**

| Plasmid replicon types | Overall n = 117 (%) | Ground turkey n = 72 (%) | Chicken breast n = 27 (%) | Pork chop n = 18 (%) |
|---|---|---|---|---|
| IncFIB(AP001918) | 73 (62.4) | 51 (70.8) | 18 (66.7) | 3 (16.7) |
| Col(MG828) | 50 (42.4) | 35 (48.6) | 12 (44.4) | 3 (16.7) |
| IncFIC(FII) | 43 (36.4) | 32 (44.4) | 10 (37) | 1 (5.6) |
| Col156 | 37 (31.4) | 23 (31.9) | 9 (33.3) | 5 (27.8) |
| IncI1 | 31 (26.3) | 25 (34.7) | 4 (14.8) | 2 (11.1) |
| p0111 | 26 (22) | 17 (23.6) | 7 (25.9) | 2 (11.1) |
| IncFII | 25 (21.2) | 14 (19.4) | 8 (29.6) | 3 (16.7) |
| IncHI2 | 18 (15.3) | 17 (23.6) | 1 (3.7) | 0 (0) |
| IncHI2A | 18 (15.3) | 17 (23.6) | 1 (3.7) | 0 (0) |
| Col440I | 17 (14.4) | 9 (12.5) | 3 (11.1) | 5 (27.8) |
| IncFIA | 15 (12.7) | 9 (12.5) | 5 (18.5) | 1 (5.6) |
| IncFIA(HI1) | 13 (11) | 11 (15.3) | 0 (0) | 2 (11.1) |
| IncFII(pSE11) | 11(9.3) | 8 (11.1) | 3 (11.1) | 0 (0) |
| IncFII(pRSB107) | 11(9.3) | 10 (13.9) | 1 (3.7) | 0 (0) |
| ColRNAI | 10 (8.5) | 6 (8.3) | 4 (14.8) | 0 (0) |
| IncX1 | 9 (7.6) | 4 (5.6) | 3 (11.1) | 2 (11.1) |
| IncHI1B(R27) | 9 (7.6) | 9 (12.5) | 0 (0) | 0 (0) |
| IncHI1A | 9 (7.6) | 9 (12.5) | 0 (0) | 0 (0) |
| ColpVC | 8 (6.8) | 2 (2.8) | 5 (18.5) | 1 (5.6) |
| IncX4 | 8 (6.8) | 8 (11.1) | 0 (0) | 0 (0) |
| IncFII(pCoo) | 8 (6.8) | 4 (5.6) | 2 (7.4) | 2 (11.1) |
| IncY | 6 (5.1) | 5 (6.9) | 1 (3.7) | 0 (0) |
| IncB/O/K/Z | 5 (4.2) | 0 (0) | 0 (0) | 5 (27.8) |
| IncHI1B(CIT) | 5 (4.2) | 5 (6.9) | 0 (0) | 0 (0) |
| IncF(29) | 5 (4.2) | 2 (2.8) | 1 (3.7) | 2 (11.1) |
| IncFIB(K) | 5 (4.2) | 1 (1.4) | 0 (0) | 4 (22.2) |
| IncFII(pHN7A8) | 5 (4.2) | 3 (4.2) | 2 (7.4) | 0 (0) |
| Col8282 | 4 (3.4) | 2 (2.8) | 2 (7.4) | 0 (0) |
| IncFIB(pB171) | 2 (1.7) | 2 (2.8) | 0 (0) | 0 (0) |
| IncX2 | 1 (0.8) | 1 (1.4) | 0 (0) | 1 (5.6) |
| ColE10 | 1 (0.8) | 0 (0) | 0 (0) | 1 (5.6) |
| IncN | 1 (0.8) | 0 (0) | 0 (0) | 1 (5.6) |
| Col(Ye4449) | 1 (0.8) | 0 (0) | 0 (0) | 1 (5.6) |
| IncA/C2 | 1 (0.8) | 1 (1.4) | 0 (0) | 0 (0) |

Seventeen isolates (14.5%) carried AMR genes on plasmid replicons identified on the same assembly scaffold, out of which 88.2% (15/17) were recovered from ground turkey and 11.8% (2/15) from chicken breast. The most prevalent replicon types were IncHI2 and IncHI2A harboring aminoglycoside resistance gene *aph(3")-Ib* in eight isolates (47.1%) and in combination with *aph(6)-Id* in three of the eight isolates recovered from ground turkey. Other AMR genes carried on plasmids include *bla*TEM-1A on IncI-1; *qacE + sul1* on IncFIC (FII); *qnrB19* on Col440I amongst others (Table 5).

## Discussion

In this study, we assessed the current prevalence and describe the characteristics of clonal types and AMR in *E. coli* isolates from retail meat, including chicken breast, ground turkey

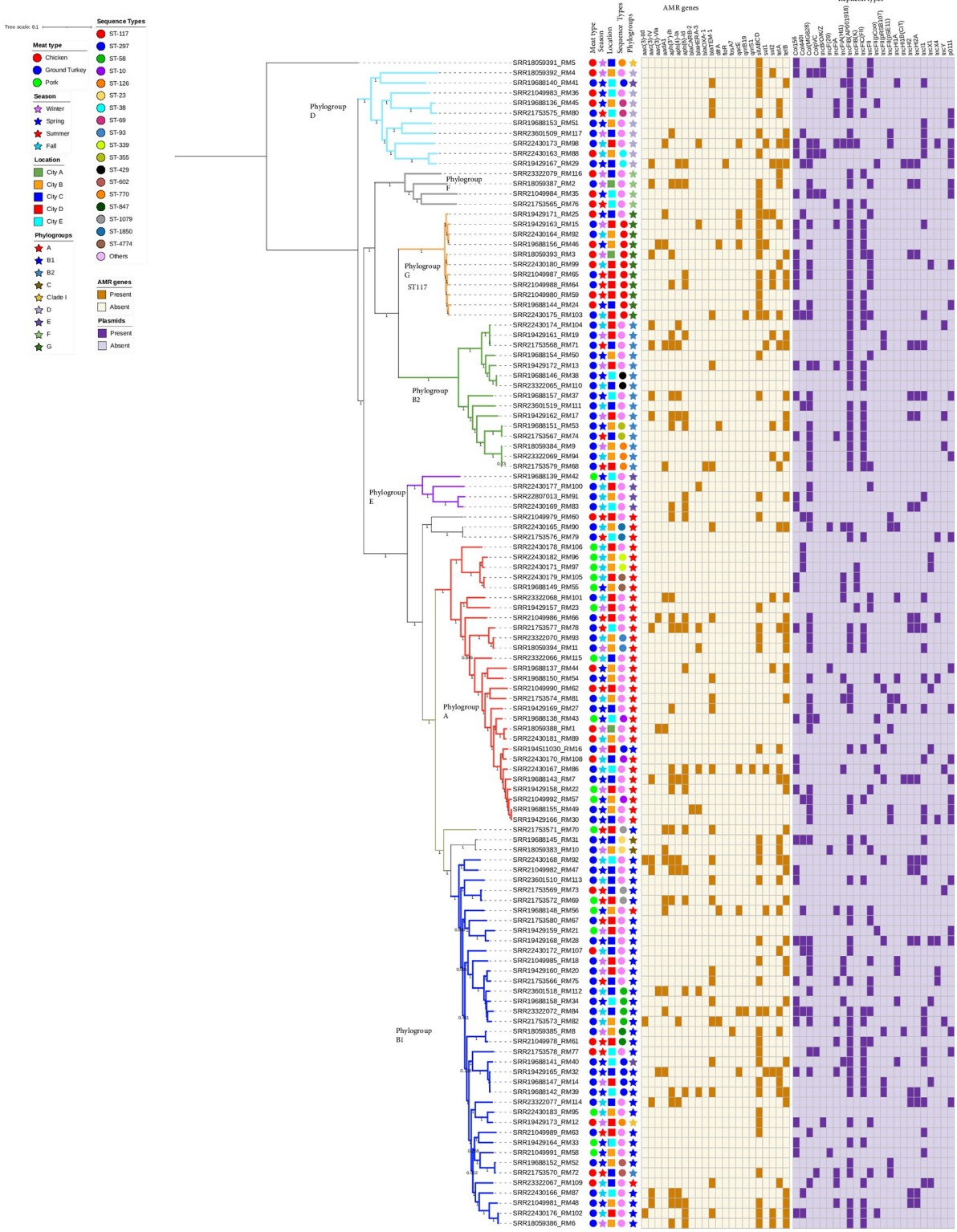

**Fig 4. SNP-based phylogeny of *Escherichia coli* isolates from retail meat in North Carolina, USA, 2022.** SNP-based maximum likelihood phylogeny of E. coli isolates visualized in interactive Tree of Life tool (iTOL). The tree was rooted in a reference *E. coli* strain K-12 MG1655. Clustering of isolates was found to be following the core genome and SNP-based phylogenies as clustering of isolates with the same sequence types and phylogroups was consistent. Shown for each isolate is the meat type, season, phylogroups, sequence types, AMR genes (brown) and plasmids (purple).

**Table 4. Clonal relationship between *Escherichia coli* isolates from different sources.**

| Clonal relationship | Sample ID | Retail meat sources | SNP Difference |
|---|---|---|---|
| A | RM68 and RM94 | Ground Turkey | 13 |
| B | RM9 and RM68 | Ground Turkey | 14 |
| C | RM9 and RM94 | Ground Turkey | 17 |
| D | RM105 and RM55 | Pork chop | 18 |
| E | RM38 and RM110 | Ground Turkey | 20 |

and pork chops purchased from retail grocery stores in North Carolina, USA. Using labeling claims, we were able to assess the role of antibiotic use on the occurrence and characteristics of AMR. We found the presence of AMR genes against different antimicrobial drug classes (aminoglycosides, tetracyclines, beta-lactamases, folate pathway antagonists, quinolones, and macrolides) in *E. coli* isolates from chicken, turkey and pork sources, but the prevalence differed by the various retail meat types. Our study results are consistent with historical genotypic prevalence reported in previous NARMS surveillance program [16–18].

In the present study, the highest prevalence of *E. coli* in the retail meat types was observed in May (spring). Our results show that there was no statistically significant association between the season of retail meat purchase and the detection of *E. coli*, although *E. coli* prevalence did differ by the month of sampling. The role of season on the presence of *E. coli* in retail meats is unclear as reports of another NARMS study in California found a prevalence of 36.7% in spring when compared to winter [17], while a similar study done in Canada found no definite seasonal trend [27]. The interplay between the effect of season and the prevalence of *E. coli* in retail meat is not fully understood [27].

**Table 5. Antimicrobial resistance (AMR) genes carried on plasmid replicon detected in 17 *Escherichia coli* isolates from retail meat, North Carolina, USA, 2022.**

| SRR# | Retail Meat types | Sequence Types | Plasmid replicon | AMR genes harbored on plasmids |
|---|---|---|---|---|
| SRR19429162 | Ground Turkey | ST141 | IncHI2, IncHI2A | *aph(3")-Ib + aph(6)-Id* |
| SRR19429165 | Ground Turkey | ST297 | IncI-1 | *bla*TEM-1A |
| SRR19688157 | Ground Turkey | ST3672 | IncHI2, IncHI2A | *aph(3")-Ib + aph(3")-Ib* |
| SRR19688156 | Chicken breast | ST117 | IncFIC(FII) | *qacE + sul1* |
| SRR21049981 | Ground Turkey | ST162 | IncHI2, IncHI2A | *aph(3")-Ib + aph(6)-Id* |
| SRR21049979 | Chicken breast | ST1771 | IncFII(pSE11) | *aph(3")-Ib + aph(6)-Id* |
| SRR21753579 | Ground Turkey | ST126 | IncFII | *bla*TEM-1A |
| SRR21753568 | Ground Turkey | ST11991 | IncHI2, IncHI2A | *aph(3")-Ib + aph(3")-Ib* |
| SRR21753577 | Ground Turkey | ST2253 | IncHI2, IncHI2A | *aph(3")-Ib + aph(6)-Id* |
| SRR23322072 | Ground Turkey | ST58 | Col440I | *qnrB19* |
| SRR22430168 | Ground Turkey | ST351 | IncHI2, IncHI2A | *aph(3")-Ib + aph(3")-Ib* |
| SRR22430167* | Ground Turkey | ST13930 | IncX2 | *aph(3')-Ia* |
| SRR22430167* | Ground Turkey | ST13930 | IncY | *aph(3")-Ib + aph(6)-Id* |
| SRR22430166 | Ground Turkey | ST4038 | IncHI2, IncHI2A | *aph(3")-Ib + aph(3")-Ib* |
| SRR22430173 | Ground Turkey | ST1938 | IncX4 | *bla*TEM-141 + *bla*TEM-206 |
| SRR22430176 | Ground Turkey | ST12733 | IncFII(pCoo) | *aac(3)-IId + bla*TEM-1B |
| SRR23601518* | Ground Turkey | ST58 | IncFII | *bla*HERA-3 |
| SRR23601518* | Ground Turkey | ST58 | IncA/C2 | *aadA1 + aac(3')-VIa* |
| SRR23322077 | Ground Turkey | ST14287 | IncHI2, IncHI2A | *aph(3")-Ib + aph(3")-Ib* |

* *E. coli* isolate with the same SRR#.

Across all isolates from all retail meat sources, we found that there was a high proportion of genotypic AMR (77.8%) among *E. coli* and a moderate occurrence of MDR *E. coli* (35.9%); this prevalence is consistent with previous data collected from retail meat in the United States over the last decade [9, 17, 19, 28, 29]. The prevalence of resistance was highest in *E. coli* from retail ground turkey when compared to other meat types. This also is consistent with findings from similar studies in Arizona which reported a high resistance prevalence of *E. coli* (90.7%) in retail turkey [19] and California reporting a prevalence of 70.4% [17]. This suggests there are differences in the management and production of turkey which may select for antimicrobial resistance or allow its persistence. It is important to note that producers in the United States are not required to make public reports on antimicrobial use data, hence it is difficult to link on-farm use of antimicrobials to development of AMR. Although between 2013 and 2017, the largest US turkey production corporations significantly reduced their overall use of antimicrobials attributed to the full implementation of FDA guidance for industry (GFI) #213, and improved antimicrobial stewardship amongst other factors [30].

The prevalence of genotypic resistance for ampicillin detected in *E. coli* from all meat types (31.6%; 37/117) was higher than the national average (11.6%; 54/466) for 2022; however, the prevalence of genotypic resistance for tetracycline was not different in *E. coli* from ground turkey (68.1%; 49/72) when compared to the national average of 62.5% (619/990). This was lower for chicken (25.9%; 7/27) and pork (22.2%; 4/18) when compared to the national average of 31.5% (147/466) and 50.4% (192/381) respectively [9]. In addition, to the ampicillin and tetracycline resistance genes, our results also show sulfonamide resistance genes and plasmid mediated quinolone resistance genes detected in retail meat. This finding aligns with past reports of the NARMS surveillance program [16–18].

Our data indicated that retail ground turkey may serve as an important source of MDR *E. coli* with the potential for transmission to humans if proper food handling practices are not used. However, there was no significant association between MDR *E. coli*, and the different antibiotics-use claims on the meat package labels. High resistance rates observed among *E. coli* isolates from turkey with "no antibiotic-use claims" may indicate that they were raised using conventional methods, however, this cannot be confirmed. This is consistent with the reports of a similar study in Arizona, USA [19]. In conventional turkey production, while there may be a greater potential for antimicrobial use when compared to "no antibiotics ever" and "organic" production, we are not able to definitively determine the use.

The *E. coli* isolates from retail meat samples displayed diverse STs and phylogroups suggesting the possibility that the isolates had evolved overtime from different *E. coli* clones. The most prevalent known STs among the isolates were ST117 (recovered from chicken and turkey), ST297 and ST58 (both recovered from chicken breast). ST10 was also detected in isolates from turkey and pork. A study conducted in California, USA in 2018 identified ST117 and ST10 in *E. coli* isolates from retail meat samples and humans [31]. Our results are consistent with reports of another related study that detected ST117 in *E. coli* recovered from commercial chicken and turkey [32] but at variance with others [32, 33] that reported ST58 and ST10 were dominant in turkey clinical *E. coli* isolates. The most dominant phylogroups in the present study were phylogroups A, B1 and B2 which was detected in isolates from all retail meat types and consistent with the literature [18, 34, 35].

The different replicon types along with the numerous *E. coli* STs in the present study highlight the diversity and complex nature of these indicator organisms. The most prevalent replicon types observed were IncFIB(AP001918), Col(MG828), IncFIC(FII), Col156, IncI1, p0111, IncFII, IncHI2, IncHI2A, Col440I and IncFIA. Seventeen isolates carried AMR genes on plasmid replicon identified on the same assembly scaffold. Replicon typing showed IncHI2 and IncHI2A harbored AMR gene *aph(3")-Ib* + *aph(6)-Id* in some isolates recovered from ground

turkey. Interestingly, the IncFIC(FII) plasmid was observed to harbor the *qacE* and *sul1* AMR gene on the assembly scaffold originating from one *E. coli* ST117 strain, recovered from retail chicken meat in the present study. This is of concern as ST117 is a known pathogenic *E. coli* lineage that has been implicated in both human and animal diseases [31, 36]. Other AMR genes carried on plasmid contigs in our study include *bla*TEM-1A on IncI-1; *qacE* + *sul1* on IncFIC (FII); *qnrB19* on Col440I amongst others. Other studies have reported *E. coli* isolates from food animals carrying the plasmid mediated quinolone resistance gene *qnrB* on a Col440I replicon carrying plasmid contig supporting our study results [37, 38].

Our analysis showed that F18:A-:B1 was the most commonly detected plasmid replicon sequence type and originated mostly from ground turkey meat products. Although these sequence types are associated with pathogenic *E. coli* causing diseases in poultry, they have been detected in faecal *E. coli* originating from healthy poultry [39, 40]. Furthermore, this study shows that ColV plasmids were harbored in *E. coli* isolated from retail meat especially ground turkey and chicken. This is not surprising because other studies have reported that ceacal *E. coli* recovered from healthy poultry were observed to carry ColV plasmids [39, 40]. In addition, avian-associated ColV plasmids have been recovered from *E. coli* originating from poultry meat products further supporting our claims [23].

Studies have reported a high diversity between *E. coli* isolates recovered from retail meat types as evident by the wide variety of STs observed [31, 41]. The *E. coli* isolated from retail meat in the present study were genetically diverse hence we did not detect any clonal relationship between isolates from the different meat types. However, our results show that isolates from the same meat type i.e., ground turkey were found to be closely related within 20 SNPs.

This study is not without limitations, particularly the small number of *E. coli* isolates characterized by sequencing may not fully encompass the diversity of *E. coli* strains recovered from retail meats, although it is a representative random sampling. Another limitation of retail level surveillance used in this study is the inconsistent or limited availability of metadata for each sample. This impacts the ability to assess factors along the food production chain from farm-to-fork that can potentially contribute to AMR and MDR in *E. coli* recovered from retail meat products. Nevertheless, meat type, label claim, packaging and season only accounted for a small portion of the variability in AMR genetic determinants among the *E. coli* isolates in this investigation, which were genotypically heterogeneous with regards to AMR.

## Conclusions

In conclusion, we detected MDR *E. coli* from retail meat types in North Carolina. Our results showed that > 40% of the retail meats purchased from grocery stores were contaminated with *E. coli*, and of these the vast majority were resistant to aminoglycosides (*aph(3")-Ib & aph(6)-Id)*, tetracycline (*tetA & tetB*) or ampicillin (*bla*TEM-1). The resistance prevalence was highest among *E. coli* isolates from turkey for most AMR genes detected. Our data indicated that ground turkey may serve as an important source of MDR *E. coli*. The isolates were diverse as only ten showed clonal relationships with a pairwise difference of ≤ 20 SNPs.

Ten of the *E. coli* isolated from retail meat had ST117 which is an emerging sequence type implicated as a human pathogen. It is important to emphasize that the bacteria isolated from retail meat samples in the present study were generic *E. coli* hence the risk of impacting negatively on human health is quite low. However, we cannot rule out the possibility of horizontal gene transfer between *E. coli* strains because our results show that plasmids, which are mobile genetic elements harbored some AMR genes which should be a source of concern for human health. Therefore, surveillance of these indicator bacteria strains should continue to serve as a warning for preventing the spread of AMR along the food chain. To further understand the

transmission dynamics, additional studies are required especially due to the persistence of these acquired AMR genes and plasmid replicon types in different *E. coli* STs.

## Supporting information

**S1 File. This file contains the metadata, resistance genes and plasmids data for the retail meat isolates.** The metadata comprises the NCBI sequence SRR, sampling month, location, meat type, meat cut, raising claim, process date, packaging type and collection year.
(XLSX)

**S1 Table. Distribution of F plasmid replicon sequence types in *E. coli* isolated from retail meat products, North Carolina.**
(DOCX)

**S2 Table. Plasmid profiling of ColV plasmid replicons in *E. coli* isolates recovered from retail meat products, North Carolina.**
(XLSX)

## Acknowledgments

The authors acknowledge the members of Thakur Lab and Jacob Lab for contributing to sample collection and processing.

## Author Contributions

**Conceptualization:** Siddhartha Thakur, Megan Jacob.

**Data curation:** Mabel Kamweli Aworh, Catherine Gensler, Erin Harrell, Lyndy Harden.

**Formal analysis:** Mabel Kamweli Aworh, Erin Harrell, Paula J. Fedorka-Cray, Megan Jacob.

**Funding acquisition:** Siddhartha Thakur.

**Investigation:** Mabel Kamweli Aworh, Siddhartha Thakur, Lyndy Harden, Paula J. Fedorka-Cray, Megan Jacob.

**Methodology:** Mabel Kamweli Aworh, Catherine Gensler, Erin Harrell, Lyndy Harden, Megan Jacob.

**Project administration:** Siddhartha Thakur, Erin Harrell, Lyndy Harden, Megan Jacob.

**Resources:** Megan Jacob.

**Supervision:** Paula J. Fedorka-Cray, Megan Jacob.

**Visualization:** Catherine Gensler, Megan Jacob.

**Writing – original draft:** Mabel Kamweli Aworh.

**Writing – review & editing:** Mabel Kamweli Aworh, Siddhartha Thakur, Catherine Gensler, Erin Harrell, Lyndy Harden, Paula J. Fedorka-Cray, Megan Jacob.

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
