## [Decision Letter · Decision Letter 0]

9 Nov 2023

PONE-D-23-34924Characteristics of antimicrobial resistance in Escherichia coli isolated from retail meat products in North CarolinaPLOS ONE

Dear Dr. Jacob,

Thank you for submitting your manuscript to PLOS ONE. After careful consideration, we feel that it has merit but does not fully meet PLOS ONE’s publication criteria as it currently stands. Therefore, we invite you to submit a revised version of the manuscript that addresses the points raised during the review process.

We look forward to receiving your revised manuscript.

Kind regards,

Marwan Osman

Academic Editor

PLOS ONE

Journal Requirements:

   "FDA National Antimicrobial Resistance Monitoring System  – ST Grant number 1U01FD007145-01 and GenomeTrakr program – Grant number 1U18FD00678801"

Reviewers' comments:

Reviewer's Responses to Questions

**Comments to the Author**

1. Is the manuscript technically sound, and do the data support the conclusions?

Reviewer #1: Yes

Reviewer #2: Yes

2. Has the statistical analysis been performed appropriately and rigorously? 

Reviewer #1: I Don't Know

Reviewer #2: I Don't Know

3. Have the authors made all data underlying the findings in their manuscript fully available?

Reviewer #1: Yes

Reviewer #2: Yes

4. Is the manuscript presented in an intelligible fashion and written in standard English?

Reviewer #1: Yes

Reviewer #2: Yes

5. Review Comments to the Author

Reviewer #1: This is a well written manuscript that presents data on an important topic. However, the manuscript would be improved in a few minor amendments could be made.

1. Can the authors provide a Table of F Plasmid Replicon Sequence Types e.g. like F18:B1 -simply reporting F types like IncF1B is not very informative. The field is improving as result of whole genome sequencing. This state of the art methodology provide great data granularity. Its a simple fix to add these analyses to existing analysis pipelines so that its collected routinely.

2. It would also be ideal - although somewhat out side the scope of the manuscript to include an estimation of ColV plasmid carriage. These plasmids are known to carry the AMR gene cargo reported in this manuscript. Its likely to be high in this collection given the nature of the samples (poultry and turkey) and you report ST117, ST58, known carriers of these important plasmids. See paper by Liu et al., 2018 for criteria Liu, C. M. et al. Escherichia coli ST131-H22 as a Foodborne Uropathogen. MBio 9, 1–11 (2018). There is a straightforward criterion used to estimate carriage of these plasmids in this paper.

Reviewer #2: This manuscript characterizes antimicrobial resistance (AMR) in E. coli from retail meat in North Carolina based on samples collected in North Carolina in 2022 by the FDA National Antimicrobial Resistance Monitoring System (NARMS) retail food surveillance program. The researchers used whole genome sequencing results to characterize AMR plus MLST and phylogroups and phylogeny based on SNP. The manuscript is technically sound, and results contribute to the research and surveillance of AMR of bacteria in retail meat.

It would be interesting to know if 2022 is the first year that the authors started the NARMS retail meat surveillance in North Carolina. If the NARMS retail food surveillance in NC started before 2022, are there any observed changes from previous years’ results?

Line: 38-40: were the differences in prevalence of E. coli among meat types statistical significantly or not?

Line 140: within how many hours post purchasing the samples were processed?

Line 226 -231: would the paragraph starting with “The accession numbers…” fit better to the section of WGS?

It would be easier for reading and catching the results if subsection headers were added under the Results section.

Line 255: what happened to the sequences of not good quality isolates, they were not submitted to NCBI?

Table 2: “Ground Turkey (antibiotic use claims)” were used twice in the first column, which is confusing and hard to tell different purposes, additional information to the table would be helpful.

Lines 369-375: what methods used to determine the AMR genes harbored on plasmid replicons? Was it based on Figure 4?

Table 5: is the table a summary of AMR genes harbored on plasmids in Figure 4 or additional analysis?

When abbreviations (e.g. MDR, GFI, IRS, SB) first appear in the text, normally a full name is provided.

The results of sequence types (ST) indicated E. coli evolvement and clonal relationships between isolates from different meat types. Were any observations in the study about the emerging STs associated with any AMR trends? It will be great to indicate in results and discussion if any of such observations in the study.

Lines 495-499: maybe the authors can check in their notice of award and determine if any standard language about how to acknowledge the NARMS grants need to be included and any other special terms. The grant was from FDA NARMS, however, the URL provided in the section is CDC’s webpage about NARMS. FDA NARMS has its’ own URL.

The resolution of Figure 4 is very good, in contrast, the resolutions of Figures 1, 2, and 3 were low and can be improved.

6. PLOS authors have the option to publish the peer review history of their article (what does this mean?). If published, this will include your full peer review and any attached files.

Reviewer #1: **Yes: **Steven Djordjevic

Reviewer #2: No

---

## [Author Response · Author response to Decision Letter 0]

28 Nov 2023

We kindly thank the reviewers for their contributions in improving our manuscript. All the comments and corrections have been addressed and documented below. 

1.Journal Requirements

Many thanks for your feedback. This has been addressed and the manuscript has been revised to ensure it meets PLOS ONE's style requirements.

2.Please note that funding information should not appear in any section or other areas of your manuscript. We will only publish funding information present in the Funding Statement section of the online submission form. Please remove any funding-related text from the manuscript. 

The funding information has been removed from the revised manuscript.

3.Thank you for stating the following financial disclosure: 

 "FDA National Antimicrobial Resistance Monitoring System – ST Grant number 1U01FD007145-01 and GenomeTrakr program – Grant number 1U18FD00678801"

 This addition has been made. 

4. Please include captions for your Supporting Information files at the end of your manuscript, and update any in-text citations to match accordingly. The captions for the Supporting Information files have been included at the end of the manuscript, and we have updated the in-text citations to match this accordingly.

5. Please review your reference list to ensure that it is complete and correct. If you have cited papers that have been retracted, please include the rationale for doing so in the manuscript text or remove these references and replace them with relevant current references. Any changes to the reference list should be mentioned in the rebuttal letter that accompanies your revised manuscript. If you need to cite a retracted article, indicate the article’s retracted status in the References list and also include a citation and full reference for the retraction notice.

 The reference list has been reviewed and new references included have been highlighted in yellow.

6. Reviewer #1: 

This is a well written manuscript that presents data on an important topic. However, the manuscript would be improved in a few minor amendments could be made. 

We appreciate the valuable comments made by the reviewer. All suggested revisions have been addressed as suggested by the reviewer. The revisions are highlighted in yellow color in the revised manuscript.

1. Can the authors provide a Table of F Plasmid Replicon Sequence Types e.g. like F18:B1 -simply reporting F types like IncF1B is not very informative. The field is improving as result of whole genome sequencing. This state of the art methodology provide great data granularity. Its a simple fix to add these analyses to existing analysis pipelines so that its collected routinely. 

This is well received and has been addressed under the Methods, Results and discussion sections. 

The methods section now reads: 

The CGE pMLST 2.0 tool (database version 2023-04-24) set at a 95% minimum identity and based on IncF-RST configuration was used for in silico plasmid MLST typing to determine the plasmid replicon sequence types [23]. 

The results section now reads: 

Of 117 E. coli isolates, only 92 (78.6%) had 50 known F plasmid replicon sequence types. The most common were F18:A-:B1 (12/92, 13%); F18:A-:B- (6/92, 6.5%); F4:A-:B20 (6/92, 6.5%); F18:A27:B1 (5/92, 5.4%); F24:A-:B1 (4/92, 4.3%); F18:A-:B20 (3/92, 3.3%); F2:A-:B- (3/92, 3.3%) and F4:A-:B1 (3/92, 3.3%) as shown in S1 Table 1. The F18:A-:B1 sequence type originated mostly from turkey (8/12) and chicken (4/12) meat products. Eighteen of the isolates did not have any F plasmid replicons while the remaining seven isolates were not typable due to the presence of novel alleles. The prevalent F plasmid replicon sequence types observed in E. coli ST117 (n=10) were F18:A-:B1 (4/10) of which most were from chicken (3/4) and F24:A-:B1 (3/10) from ground turkey (2/3). Eighteen of the isolates did not have any F plasmid replicons while the remaining seven isolates were not typable due to the presence of novel alleles.

The discussion section now reads: 

Our analysis showed that F18:A-:B1 was the most commonly detected plasmid replicon sequence type and originated mostly from ground turkey meat products. Although these sequence types are associated with pathogenic E. coli causing diseases in poultry, they have been detected in faecal E. coli originating from healthy poultry [39, 40].

8. 2. It would also be ideal - although somewhat out side the scope of the manuscript to include an estimation of ColV plasmid carriage. These plasmids are known to carry the AMR gene cargo reported in this manuscript. Its likely to be high in this collection given the nature of the samples (poultry and turkey) and you report ST117, ST58, known carriers of these important plasmids. See paper by Liu et al., 2018 for criteria Liu, C. M. et al. Escherichia coli ST131-H22 as a Foodborne Uropathogen. MBio 9, 1–11 (2018). There is a straightforward criterion used to estimate carriage of these plasmids in this paper.

 This is well received and has been addressed under the methods, results and discussion sections. 

The methods section now reads: 

From the output generated, if an isolate had at least one gene from either of the following sets: (i) virulence factors: ompT and hlyF, and (ii) resistance genes: sitABCD, it was deemed ColV plasmid positive [24]. 

The results section now reads: 

Plasmid profiling identified presumptive ColV positive plasmid replicons in 72.6% (85/117) of E. coli isolates recovered from different retail meat products (S2 Table). Of these, 67.1% (57/85) were recovered from ground turkey, chicken 25.9% (22/85) and pork 7.1% (6/85).

The discussion section now reads:

Furthermore, this study shows that ColV plasmids were harbored in E. coli isolated from retail meat especially ground turkey and chicken. This is not surprising because other studies have reported that caecal E. coli recovered from healthy poultry were observed to carry ColV plasmids [39, 40]. In addition, avian-associated ColV plasmids have been recovered from E. coli originating from poultry meat products further supporting our claims [23].

9. Reviewer #2: 

This manuscript characterizes antimicrobial resistance (AMR) in E. coli from retail meat in North Carolina based on samples collected in North Carolina in 2022 by the FDA National Antimicrobial Resistance Monitoring System (NARMS) retail food surveillance program. The researchers used whole genome sequencing results to characterize AMR plus MLST and phylogroups and phylogeny based on SNP. The manuscript is technically sound, and results contribute to the research and surveillance of AMR of bacteria in retail meat.

Thank you for your positive feedback. 

10. It would be interesting to know if 2022 is the first year that the authors started the NARMS retail meat surveillance in North Carolina. If the NARMS retail food surveillance in NC started before 2022, are there any observed changes from previous years’ results? 

Excellent question and we agree. Although it is not the first year of data collection in North Carolina, the other data is not currently published. We do hope that comparisons can be made in the future. 

11. Line: 38-40: were the differences in prevalence of E. coli among meat types statistical significantly or not? 

There were statistically significant differences observed in the prevalence of E. coli among meat types. This correction has been made in the Results section.

This section now reads: 

The differences observed in prevalence of E. coli among retail meat types were statistically significant (X2=13.17; p<0.01). 

12. Line 140: within how many hours post purchasing the samples were processed? 

We have now addressed this under Methods section.

This section now reads: 

On arrival at the lab, samples were refrigerated at 4°C and processed within 96 hours for detection of multiple bacteria, including E. coli. 

13. Line 226 -231: would the paragraph starting with “The accession numbers…” fit better to the section of WGS?

It would be easier for reading and catching the results if subsection headers were added under the Results section. This is well received and has been addressed under Results section.

Kindly refer to page 11, lines 274-278 of revised manuscript.

14. Line 255: what happened to the sequences of not good quality isolates, they were not submitted to NCBI? 

These poor-quality sequences data did not meet the criteria for being submitted to NCBI, hence these were excluded from the analysis. 

15. Table 2: “Ground Turkey (antibiotic use claims)” were used twice in the first column, which is confusing and hard to tell different purposes, additional information to the table would be helpful. 

This correction has been made. 

The Table legend has been updated and now reads: 

#Analysis of different antibiotics use claims among ground turkey meat products; ##The second analysis compares the difference between “no claim” versus “claim” among ground turkey meat products.

16. Lines 369-375: what methods used to determine the AMR genes harbored on plasmid replicons? Was it based on Figure 4? 

Revised for clarity under the Methods section.

The method section now reads: 

The in silico analysis of acquired resistance genes and replicon typing for each E. coli isolate was used to identify AMR genes harbored on plasmid replicons and was conducted using the Mobile Element Finder tool (database version 1.0.2, 2020-06-09) accessed online via the Center for Genomic Epidemiology (CGE) website (https://cge.food.dtu.dk/services/MobileElementFinder/) [22]. 

17. Table 5: is the table a summary of AMR genes harbored on plasmids in Figure 4 or additional analysis? 

Table 5 represents additional analysis.

18. When abbreviations (e.g. MDR, GFI, IRS, SB) first appear in the text, normally a full name is provided. 

This is well received and has been addressed in the revised manuscript. However, solution-SB and solution-IRS are not abbreviations rather they are labels of solution in the QIAGEN DNA extraction kit (product names).

The abstract now reads: 

Genes associated with AMR were detected in 77.8% (91/117) of the isolates and 35.9% (42/117) were defined as multidrug resistant (MDR: being resistant to ≥3 distinct classes of antimicrobials). 

The discussion section now reads: 

Although between 2013 and 2017, the largest US turkey production corporations significantly reduced their overall use of antimicrobials attributed to the full implementation of FDA guidance for industry (GFI) #213, and improved antimicrobial stewardship amongst other factors [30].

19. The results of sequence types (ST) indicated E. coli evolvement and clonal relationships between isolates from different meat types. Were any observations in the study about the emerging STs associated with any AMR trends? It will be great to indicate in results and discussion if any of such observations in the study.

 No, the observations in this study were not about emerging STs associated with any AMR trends, rather an inclusive surveillance of all isolates. It is important to note that the clonal relationship observed in this study were between isolates originating from the same types of retail meat products. 

20. Lines 495-499: maybe the authors can check in their notice of award and determine if any standard language about how to acknowledge the NARMS grants need to be included and any other special terms. The grant was from FDA NARMS, however, the URL provided in the section is CDC’s webpage about NARMS. FDA NARMS has its’ own URL. 

This is well received. The funders’ information has been removed from the revised manuscript text based on Journal requirements. 

21. The resolution of Figure 4 is very good, in contrast, the resolutions of Figures 1, 2, and 3 were low and can be improved. This is well received and has been addressed. 

Figures 1, 2 and 3 have been improved.

---

## [Decision Letter · Decision Letter 1]

18 Dec 2023

Characteristics of antimicrobial resistance in Escherichia coli isolated from retail meat products in North Carolina

PONE-D-23-34924R1

Dear Dr. Jacob,

We’re pleased to inform you that your manuscript has been judged scientifically suitable for publication and will be formally accepted for publication once it meets all outstanding technical requirements.

Kind regards,

Marwan Osman

Academic Editor

PLOS ONE

Reviewers' comments:

Reviewer's Responses to Questions

**Comments to the Author**

1. If the authors have adequately addressed your comments raised in a previous round of review and you feel that this manuscript is now acceptable for publication, you may indicate that here to bypass the “Comments to the Author” section, enter your conflict of interest statement in the “Confidential to Editor” section, and submit your "Accept" recommendation.

Reviewer #1: All comments have been addressed

Reviewer #2: All comments have been addressed

2. Is the manuscript technically sound, and do the data support the conclusions?

Reviewer #1: Yes

Reviewer #2: Yes

3. Has the statistical analysis been performed appropriately and rigorously? 

Reviewer #1: Yes

Reviewer #2: Yes

4. Have the authors made all data underlying the findings in their manuscript fully available?

Reviewer #1: Yes

Reviewer #2: Yes

5. Is the manuscript presented in an intelligible fashion and written in standard English?

Reviewer #1: Yes

Reviewer #2: Yes

6. Review Comments to the Author

Reviewer #1: The revised manuscript has addressed all my concerns. This is a relatively small but important contribution _ I hope the authors maintain their interest in conducting genomic surveillance studies in the future.

Reviewer #2: (No Response)

7. PLOS authors have the option to publish the peer review history of their article (what does this mean?). If published, this will include your full peer review and any attached files.

Reviewer #1: **Yes: **Steven P. Djordjevic

Reviewer #2: No

---

## [Editor Report · Acceptance letter]

26 Dec 2023

PONE-D-23-34924R1 

PLOS ONE

Dear Dr. Jacob, 

I'm pleased to inform you that your manuscript has been deemed suitable for publication in PLOS ONE. Congratulations! Your manuscript is now being handed over to our production team.

Kind regards, 

on behalf of

Dr. Marwan Osman 

Academic Editor

PLOS ONE